# Use of common blood parameters for the differential diagnosis of childhood infections

**Weiying Wang** **\*, Shu Hua Li**

Department of Pediatric Outpatient, Guangzhou Women and Children's Medical Center, Guangzhou, China

\* xiaobaihuaqimei@sina.com

## Abstract

### Background

Routine laboratory investigations are not rapidly available to assist clinicians in the diagnosis of pediatric acute infections. Our objective was to evaluate some common blood parameters and use them for the differential diagnosis of childhood infections.

### Methods

This retrospective study was conducted between October 2019 and September 2020 at Guangzhou Women and Children's Medical Center, China. We performed blood tests in patients infected with DNA viruses (n = 402), RNA viruses (n = 602), gram-positive organisms (G+; n = 421), gram-negative organisms (G−; n = 613), or *Mycoplasma pneumoniae* (n = 387), as well as in children without infection (n = 277). The diagnostic utility of blood parameters to diagnose various infections was evaluated by logistic regression analysis.

### Results

The most common G+ organism, G− organism, and virus were *Streptococcus pneumoniae* (39.7%), *Salmonella typhimurium* (18.9%), and influenza A virus (40.2%), respectively. The value of logit ($P$) = 0.003 × C-reactive protein (CRP) − 0.011 × hemoglobin (HGB) + 0.001 × platelets (PLT) was significantly different between the control, RNA virus, DNA virus, *M. pneumoniae*, G− organism, and G+ organism groups (2.46 [95% CI, 2.41–2.52], 2.60 [2.58–2.62], 2.70 [2.67–2.72], 2.78 [2.76–2.81], 2.88 [2.85–2.91], and 2.97 [2.93–3.00], respectively; p = 0.00 for all). The logistic regression-based model showed significantly greater accuracy than the best single discriminatory marker for each group (logit [$P_{infection}$] vs. CRP, 0.90 vs. 0.84, respectively; logit [$P_{RNA}$] vs. lymphocytes, 0.83 vs. 0.77, respectively; p = 0.00). The area under curve values were 0.72 (0.70–0.74) for HGB and 0.81 (0.79–0.82) for logit ($P_{virus/bacteria}$) to diagnose bacterial infections, whereas they were 0.72 (0.68–0.74) for eosinophils and 0.80 (0.78–0.82) for logit ($P_{virus/bacteria}$) to diagnose viral infections. Logit ($P_{virus/bacteria}$) < −0.45 discriminated bacterial from viral infection with 78.9% specificity and 70.7% sensitivity.

**Data Availability Statement:** All relevant data can be found at: https://doi.org/10.17026/dans-x89-68f9.

**Funding:** This research received no specific grant from any funding agency in the public, commercial or not-for-profit sectors.

**Competing interests:** The authors have declared that no competing interests exist.

## Conclusions

The combination of CRP, HGB, PLT, eosinophil, monocyte, and lymphocyte counts can distinguish between the infectious pathogens in children.

## Background

Bacterial infections are important causes of morbidity and mortality among children. It is crucial to diagnose bacterial infections and distinguish between bacterial and non-bacterial infections. Unfortunately, almost three-quarters of patients with a viral syndrome receive antibiotics [1]. Laboratory parameters, such as leukocytes or white blood cells (WBCs) and C-reactive protein (CRP), provide diagnostic information. The hepatic acute phase reactant CRP is the most commonly used biomarker of bacterial infections, which is also recommended by the febrile neutropenia guideline of the European Society of Medical Oncology (ESMO) [2]. Currently, bacterial and viral infections are mainly differentiated on the basis of WBC and CRP levels [3, 4], which typically has sensitivity of 60% and specificity of 70%, resulting in a high rate of misdiagnosis [5]. Notably, there is no standard cutoff level to diagnose bacterial infections, a low CRP level does not exclude bacterial infection, and a high CRP level can also occur in the absence of bacterial infection. This highlights the limitations of use of CRP level as an inflammatory biomarker.

In children, the neutrophil-to-lymphocyte ratio (NLR) can differentiate between viral and bacterial pneumonia [6], and is a diagnostic marker of acute appendicitis [7]. However, physical examination findings and routine laboratory investigations cannot accurately differentiate between benign viral and severe bacterial infections in children with fever [8]. New markers for bacterial infection have been discovered, including presepsin, procalcitonin, CD64, and pro-adrenomedullin (proADM). Presepsin, procalcitonin, and CD64 are diagnostic markers for severe sepsis and septic shock, whereas proADM is a prognostic marker of bacterial infections [9–12]; however, these markers cannot be used for the diagnosis of mild bacterial or viral infections [13]. Although microbiological culture is the gold standard for diagnosing bacterial infections, culturing of bacteria is time-consuming [14]. Early administration of antibiotics in bacterial infections improves the outcome and reduces the mortality among patients [15]. Therefore, the development of rapid and accurate methods of diagnosis is warranted. The aim of this study was to assess the usefulness of commonly available blood parameters and cut-off values thereof in differentiating infections due to RNA viruses, DNA viruses, *Mycoplasma pneumoniae*, gram-positive organisms (G+), and gram-negative organisms (G−) in febrile pediatric patients.

## Patients and methods

### Study population

Data were retrospectively collected from patients treated at the Guangzhou Women and Children's Medical Center, China, which is a large, tertiary care children's hospital, between October 2019 and September 2020. This study included 2,425 patients (aged ≤ 17 years) whose blood culture, polymerase chain reaction (PCR), or serological test (i.e., immunoglobulin test) suggested acute bacterial, viral, or *M. pneumoniae* infection. In addition, urine, stool, cerebrospinal fluid, or bronchoalveolar lavage fluid cultures were performed, where necessary. The PCR or immunoglobulin test tested for ten pathogens, namely Human Bocavirus (HBoV), influenza A virus (IAV), influenza B virus (IBV), parainfluenza virus (PIV), rhinovirus (RHV),

respiratory syncytial virus (RSV), adenovirus (ADV), Epstein-Barr virus (EBV), enterovirus (EV), and herpes simplex virus (HSV). Two hundred and seventy-seven children without infection were included in the control group.

## Inclusion criteria

Patients aged ≤ 17 years with suspected bacterial or viral infections were included in the study. Bacterial infections were identified by a positive bacterial blood, urine, stool, cerebrospinal fluid, or bronchoalveolar lavage fluid culture. Viral or *M. pneumoniae* infection was identified by a positive relevant PCR or serological test. In the case of multiple hospital admissions, only the first was analyzed.

## Exclusion criteria

Based on a review of medical records, we excluded patients with a positive viral test, diagnosis of a potential bacterial infection, such as cellulitis, cholecystitis, erysipelas, pneumonia, pyelonephritis, or septicemia, that suggested multi-organism infection (n = 34), ≥ 1 pathogen type (n = 23), hematological cancer with variable blood cell counts due to the cancer or chemotherapy (n = 11), or bacterial contaminants on bacterial culture (i.e., negative cultures; n = 22) (Fig 1). Contamination was defined as the presence of multiple coagulase-negative *Staphylococcus* species, *Bacillus* species, *Propionibacterium acnes*, or Corynebacterium species in a single set of blood cultures; these bacteria are frequent contaminants [16].

Patients were categorized into those with bacterial (n = 1034), viral (n = 1004), and *M. pneumoniae* (n = 387) infections. Based on the diagnostic criteria for pediatric sepsis [17], patients in the bacterial infection group were further classified into G+ (n = 421) and G– (n = 613) organism groups. Similarly, based on the classification of viruses [18], patients in the viral infection group were further classified into DNA (n = 402) and RNA (n = 602) virus groups. The control group included 277 healthy children without infections or inflammatory diseases who underwent routine health check at the study center.

## Laboratory data

The medical records were reviewed to record the medical history (sex and age) and results of the laboratory evaluation (including blood cell counts, CRP level, throat swab, bronchoalveolar lavage fluid reverse transcription (RT)-PCR, blood, urine, and stool cultures, and lumbar puncture to identify the infection source). CRP secretion is regulated by cytokines, and the CRP level reaches its peak at 48 hours [19, 20]. After the bacterial trigger for inflammation is eliminated, CRP levels decrease rapidly, with a half-life of almost 19 hours [20, 21]. Delayed normalization of CRP levels after the first 3–7 days of follow up is suggestive of inappropriate antibiotic selection [22]. Hence, we recorded the blood counts and CRP levels on days 3–7 after symptom onset. We did not evaluate the procalcitonin level because our aim was to study the commonly available diagnostic markers.

Our aim was to assess the diagnostic utility of commonly used laboratory parameters in distinguishing between certain pathogen types in pediatric patients, using the best single discriminatory marker for each infection group as a comparator. We further used logistic regression to develop models for distinguishing among the six groups.

## Statistical analyses

We compared quantitative data using Student's t-test or one-way ANOVA and compared frequencies using the chi-square test between the groups. Correlations were analyzed using

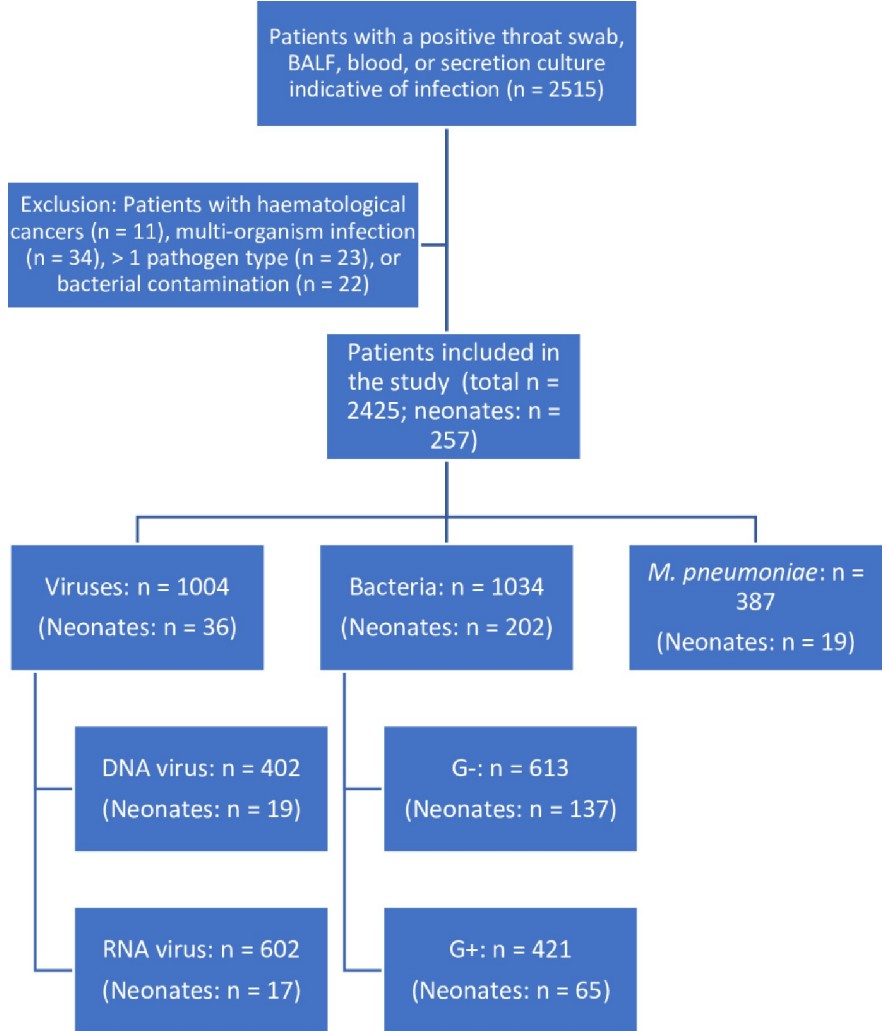

**Fig 1. Flowchart for patient selection.** The numbers of neonates are presented in parentheses. BALF, bronchoalveolar lavage fluid; *M. pneumoniae*, *Mycoplasma pneumoniae*; G+, Gram-positive organisms; G−, Gram-negative organisms.

Pearson's R and Spearman's R. After adjusting for the potential predictors, multivariate logistic regression was performed for selected data, and receiver operating characteristic (ROC) curve was constructed to calculate probabilities. To test the performance of a numerical parameter as a biomarker for classification, we used the ROC to calculate the area under curve (AUC), where the positive class was the pathogen type. The sensitivity and specificity were calculated to evaluate the diagnostic accuracy. The measurement data were expressed as mean ± standard deviation (SD) or median with interquartile range (IQR). The 95% confidence intervals (CIs) were used to quantify uncertainty. P values < 0.05 were considered statistically significant. All statistical analyses were performed using SPSS statistical software (version 21.0; IBM Corp., Armonk, NY, USA).

## Ethics statement

This study was conducted in accordance with the Declaration of Helsinki and approved by the Guangzhou Women and Children's Medical Center Ethics Committee (no.: 2020110819342581).

Patient consent or additional permission from the hospital was not required because all study data were retrospectively collected from the medical records as part of the usual clinical process.

## Results

### Demographic characteristics of patients

The bacterial infection group consisted of 1034 patients (566 males; median age [IQR]: 2.1 [0.4–3.1] years). Patients with bacterial infections were younger than those with viral or *M. pneumoniae* infections; however, the proportion of male gender was similar among the three groups. The ICU occupancy rate was higher among patients with bacterial infection than those with other infections. These analyses are presented in Table 1. The specific pathogens are reported in Table 1 and Fig 2A. A wide array of bacterial species was isolated from the cohort, with *Streptococcus* spp. (n = 167), *Enterococcus* spp. (n = 127), *Staphylococcus* spp. (n = 127), *Salmonella* spp. (116), *Klebsiella* spp. (n = 112), *Escherichia coli* (n = 73), and *Pseudomonas aeruginosa* (n = 53) most often cultured. The most common G+ organism was *Streptococcus* spp. (39.7%). *S. Typhimurium* was the most common G− organism (18.9%), followed by *Klebsiella pneumoniae* (18.3%), and *E. coli* (11.9%), consistent with the results of a previous study [23]. In the viral infection group, 40.2% had IAV infection (579 males; median age: 4.2 years). The median age of patients with *M. pneumoniae* infection (221 males) and controls (157 males) were 3.9 (IQR 1.9–5.7) and 2.5 (0.0–1.0) years, respectively. There was no statistical difference between the groups in terms of gender. Because the disease prevalence differs by age, age is considered the most important demographic characteristic [24]. Importantly, the median age was higher in the viral infection group than bacterial infection group, and the morbidity of bacterial infections was higher in neonates than older children (78.60% vs. 42.39%, respectively). Fig 2B and 2C presents the etiological distribution of infections by months, and may assist physicians to predict the prevalent pathogen in each month. As an example, up to 48.3% of patients with RNA viral infections presented in December, whereas 25.4% with DNA viral infections presented in July. Among these patients, IAV and ADV was present in 68.9% and 60.2% of the total patients in the RNA and DNA virus groups, respectively.

Patients with bacterial infections had higher levels of CRP, leukocytes, neutrophils, eosinophils, and monocytes compared with patients with viral infections. Both groups with infection showed higher CRP levels compared with controls, but the mean CRP level was higher in the bacterial group compared with the viral group (32.94 ± 43.88 vs. 13.58 ± 21.86 mg/l, respectively; $p$ = 0.00) and in the G+ organism group compared with the G− organism group (40.11 ± 42.56 vs. 28.21 ± 21.57 mg/l, respectively; $p$ = 0.00). The CRP level was significantly higher in patients with DNA virus infections compared with RNA virus infections, in line with previous studies [8]. However, the NLR was lower in patients with DNA virus infections compared with those with RNA virus and *M. pneumoniae* infections ($p$ = 0.00). Patients with RNA virus infections had significantly higher NLR compared with patients with DNA virus, *M. pneumoniae*, and bacterial infections (3.73 ± 3.18, 1.92 ± 2.34, 1.80 ± 1.53, and 2.41 ± 3.80, respectively; $p$ = 0.00) Furthermore, the bacterial group, as compared with the viral group, had lower hemoglobin (HGB) level (112.25 ± 22.24 vs. 122.38 ± 14.23 g/L, respectively; $p$ = 0.00), but higher levels of WBCs, neutrophils, and platelets (PLTs) (12.19 × $10^9$/L ± 8.92 × $10^9$/ L and 9.71 × $10^9$/L ± 5.55 × $10^9$/L; 6.41 ± 5.24 × $10^9$/L and 5.44 × $10^9$/L ± 3.61× $10^9$/L; 347.44 × $10^9$ /L ± 156.79 × $10^9$/L and 292.00 × $10^9$/L ± 118.87 × $10^9$/L, respectively; $p$ = 0.00); however, lymphocyte, eosinophil, and monocyte counts were not significantly different between the groups. The WBC and PLT counts were significantly lower in pediatric patients with RNA virus infections than other patients ($p$ = 0.00). In addition, in the general cohort of patients with acute infections, HGB level was significantly reduced (119.49 ± 17.82 vs.

**Table 1. Demographic and clinical characteristics of patients infected with RNA viruses, DNA viruses, *M. pneumoniae*, G− organisms, and G+ organisms.**

| | Total | Control | RNA virus | DNA virus | *M. pneumoniae* | G− organism | G+ organism | R | P |
|---|---|---|---|---|---|---|---|---|---|
| Male, n (%) | 1523 (58.8) | 157 (56.8) | 330 (54.9) | 249 (61.9) | 221 (57.1) | 331 (61.5) | 235 (60.6) | −0.04 | 0.14 |
| Female, n (%) | 1179 (41.2) | 120 (43.2) | 272 (45.1) | 153 (37.1) | 166 (42.9) | 282 (38.5) | 186 (39.4) | | |
| Median age, years (IQR) | 3.2 (0.5–5.0) | 2.5 (0.0–1.0) | 5.4 (2.7–8.0) | 3.1 (0.7–4.6) | 3.9 (1.9–5.7) | 1.9 (0.1–2.1) | 2.3 (0.3–3.6) | 0.02 | 0.27 |
| Neonates | 356 | 99 | 17 | 19 | 19 | 137 | 65 | | |
| Setting, n (%) | | | | | | | | 0.45 | 0 |
| ICU | 315 | 0 | 9 (1.5) | 71 (17.8) | 2 (0.5) | 136 (22.6) | 97 (23.9) | | |
| Outpatients | 1786 | 260 (100) | 558 (92.7) | 254 (63.9) | 336 (86.4) | 238 (39.6) | 172 (41.2) | | |
| Month with the highest morbidity (n, %) | 12 (519, 20.5) | _ | 12 (291, 48.3) | 7 (92, 25.4) | 7 (77, 19.7) | 10 (100, 16.4) | 11 (66, 15.8) | −0.11 | 0 |
| Most frequent pathogen (n,%) | IAV (415, 16.3%) | _ | IAV (415, 68.9%) | ADV (262, 60.2%) | _ | Salmonella Typhimurium (116, 18.9%) | Streptococcus (167, 39.7%) | 0.04 | 0.03 |
| Laboratory findings | | | | | | | | | |
| CRP | 19.94 (18.64–21.22) | 0.86 (0.62–1.11) | 10.07 (8.64–11.52) | 19.14 (16.51–21.76) | 9.86 (8.20–11.52) | 28.21 (24.86–31.57) | 40.11 (35.76–44.56) | 0.11 | 0 |
| Eosinophils | 0.27 (0.37–0.43) | 0.40 (0.37–0.43) | 0.09 (0.07–0.11) | 0.27 (0.23–0.32) | 0.24 (0.22–0.27) | 0.36 (0.32–0.40) | 0.32 (0.29–0.36) | 0.02 | 0.34 |
| Hemoglobin | 120.23 (119.46–121.00) | 136.22 (132.97–139.48) | 125.79 (124.81–126.78) | 117.25 (115.75–118.76) | 124.75 (123.56–125.94) | 112.98 (111.12–114.84) | 111.08 (109.13–113.03) | −0.18 | 0 |
| Lymphocytes | 3.97 (3.72–4.23) | 5.08 (4.87–5.28) | 2.29 (2.14–2.44) | 5.31 (3.72–6.90) | 3.73 (3.54–3.92) | 4.21 (4.00–4.46) | 4.27 (4.02–4.52) | −0.04 | 0.06 |
| Monocytes | 0.88 (0.81–0.94) | 0.81 (0.76–0.87) | 0.75 (0.72–0.78) | 0.89 (0.84–0.94) | 0.64 (0.61–0.68) | 1.11 (0.79–1.43) | 0.99 (0.93–1.05) | 0.01 | 0.62 |
| Neutrophils | 5.57 (5.41–5.73) | 3.63 (3.28–3.97) | 5.44 (5.17–5.72) | 5.44 (5.07–5.82) | 5.02 (4.74–5.30) | 5.93 (5.53–6.32) | 7.14 (6.61–7.67) | 0.09 | 0 |
| Platelets | 328.39 (323.22–333.56) | 356.64 (344.04–369.24) | 266.14 (258.21–274.07) | 330.73 (317.52–343.93) | 351.75 (339.38–364.12) | 339.06 (326.85–351.26) | 359.29 (343.92–374.66) | 0.06 | 0 |
| RBCs | 4.46 (4.40–4.53) | 4.56 (4.50–4.64) | 4.67 (4.63–4.71) | 4.45 (4.39–4.51) | 4.69 (4.64–4.75) | 4.29 (4.03–4.54) | 4.15 (4.08–4.23) | −0.02 | 0.32 |
| NLR | 2.39 (2.27–2.51) | 0.90 (0.75–1.05) | 3.73 (3.44–4.01) | 1.92 (1.71–2.13) | 1.79 (1.64–1.95) | 2.33 (1.99–2.66) | 2.56 (2.27–2.86) | 0.08 | 0 |
| WBCs | 10.68 (10.42–10.94) | 9.94 (9.53–10.34) | 8.74 (8.28–9.20) | 11.18 (10.70–11.67) | 9.67 (9.33–10.01) | 11.80 (11.00–12.60) | 12.80 (12.14–13.46) | 0.05 | 0.02 |

Abbreviations: *M. pneumoniae*, *Mycoplasma pneumoniae*; G−, gram-negative organisms; G+, gram-positive organisms; IQR, interquartile range; ICU, intensive care unit; CRP, C-reactive protein; RBCs, red blood cells; NLR, neutrophil-to-lymphocyte ratio; WBCs, white blood cells.

136.22 ± 27.38 g/L, respectively), while NLR and neutrophil counts increased, compared with controls without infections (2.52 ± 3.29 vs. 0.90 ± 1.27; 5.60 ± 4.17 vs. 3.63 ± 2.91 × 10$^9$/L, respectively; $p = 0.00$). Since the reference blood counts for neonates differ from those for older children, the results for neonates are presented separately; the results indicated higher morbidity with bacterial infections than viral infections (Table 1).

## Multivariate logistic regression analysis

The multivariate logistic regression analysis revealed significant associations of CRP, HGB, neutrophil, PLT, and WBC levels, and NLR with infections ($p < 0.05$ for all). The associations with CRP, HGB, and PLT levels remained statistically significant ($p = 0.00$) after the application

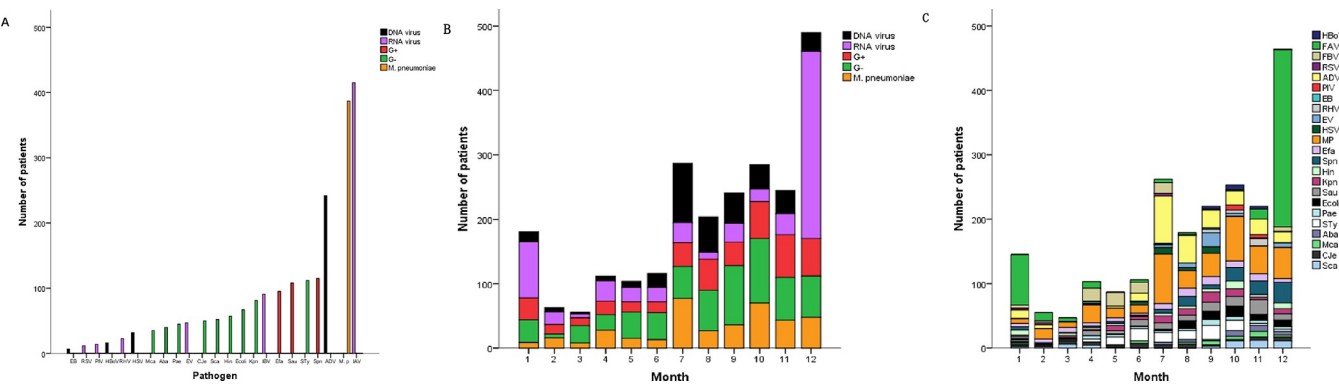

**Fig 2.** A, Number of patients infected with specific pathogens. B, Distribution of patients infected with DNA viruses, RNA viruses, G+ bacteria, G− bacteria, and *M. pneumoniae* in 12 months. C, Distribution of specific pathogens in the five groups by the 12 months. The X axis represents the months, and the Y axis represents the number of patients infected with specific pathogens. *M. pneumoniae*, *Mycoplasma pneumoniae* group; G+, gram-positive organisms group; G−, gram-negative organism group; HBoV, human Bocavirus; IAV/FAV, influenza A virus; IBV/FBV, influenza B virus; PIV, parainfluenza virus; RHV, rhinovirus; RSV, respiratory syncytial virus; ADV, adenovirus; EBV, Epstein-Barr virus; EV, enterovirus; HSV, herpes simplex virus; Efa, *Enterococcus faecium*; Hin, *Haemophilus influenzae*; Kpn, *Klebsiella pneumoniae*; Sau, *Staphylococcus aureus*; E. coli, *Escherichia coli*; Spn, *Streptococcus Peroris*; Sty, *Salmonella typhimurium*; Aba, *Acinetobacter baumannii*; Mca, *Moraxella catarrhalis*; Cje, *Campylobacter Jejuni*; Pae, *Pseudomonas aeruginosa*; Sca, Shigella.

of the forward regression model, whereas WBC and neutrophil counts, and NLR were excluded from the model. Based on the variables selected for multivariate logistic regression analysis, we developed a logistic regression-based model for distinguishing among the six groups:

Logit $(P) = 0.003 \times (CRP − 0.011) \times (HGB + 0.001) \times PLT$

The mean logit $(P)$ values were 2.46 (95% CI, 2.41–2.52), 2.60 (2.58–2.62), 2.70 (2.67–2.72), 2.78 (2.76–2.81), 2.88 (2.85–2.91), and 2.97 (2.93–3.00) for children in the control, RNA virus, DNA virus, *M. pneumoniae*, G− organism, and G+ organism groups, respectively ($p = 0.00$ for comparison between any two means). Using a combination of HGB, PLT, and CRP levels, the AUCs for predicting acute infections and infections due to RNA viruses, DNA viruses, *M. pneumoniae*, G− organisms, and G+ organisms were 0.75 (95% CI, 0.72–0.78), 0.76 (0.73–0.78), 0.52 (0.47–0.54), 0.60 (0.57–0.62), 0.65 (0.63–0.67), and 0.72 (0.69–0.74), respectively. The classification quality of the parameter for identifying DNA viruses, *M. pneumoniae*, and G − organisms (AUCs < 0.70) was unacceptable according to the criteria developed by Hosmer and Lemeshow, who suggested that AUCs of 0.70–0.80, 0.80–0.90, and ≥ 0.9 respectively offer acceptable, excellent, and outstanding discrimination abilities [25]. Furthermore, there were no significant differences in these AUCs compared with the largest AUC for a single bio-marker in these three groups (eosinophils in DNA virus group: 0.60, 95% CI 0.57–0.63; mono-cytes in *M. pneumoniae* group: 0.65, 0.62–0.68; HGB in the G− organism group: 0.65, 0.62–0.68). We addressed this limitation by developing an additional logistic regression-based model comprised of statistically significant components in each group as a supplement:

Logit $(P_{\text{infection}}) = 0.542 * (CRP − 0.05) * (HGB + 24.345 − 0.035) * (LYMPH + 7.946)$

Logit $(P_{\text{virus/bacteria}}) = −0.988 − 0.013 * (CRP − 3.457) * (EO + 0.018) * (HGB − 0.003) * PLT$

Logit $(P_{\text{RNA}}) = 0.941 − 0.031 * (CRP − 2.343) * (EO − 0.41) * LYMPH$

Logit $(P_{\text{DNA}}) = −1.809 − 2.615 * (EO + 0.089) * LYMPH$

Logit $(P_{M. pneumoniae}) = −3.117 + 0.023 * (HGB − 1.214) * (MONO − 0.192) * NLR$

Logit $(P_{\text{G−}}) = 0.861 + 1.104 * (EO − 0.02) * HGB$

Logit $(P_{\text{G+}}) = 0.012 * (CRP − 0.018) * (HGB + 0.001) * PLT + EO * 0.555$

Based on the overall study population, logit $(P_{\text{infection}})$ showed a concentration-response relationship between children with and without infections. Using data from all patients with infections, logit $(P_{\text{virus/bacteria}})$ was developed for differentiating patients with and without

bacterial infections, and showed a concentration-response relationship among children with and without viral infection. Lower logit ($P_{virus/bacteria}$) is associated with greater likelihood of bacterial infection, whereas higher logit ($P_{virus/bacteria}$) is associated with viral infections. Similarly, based on all patients with infections, logit ($P_{RNA}$), logit ($P_{DNA}$), logit ($P_{M. pneumoniae}$), logit ($P_{G-}$), and logit ($P_{G+}$) were developed for differentiating children with and without RNA virus, DNA virus, *M. pneumoniae*, G− organism, and G+ organism infections (logit [$P_{RNA}$] AUC: 0.83 [95% CI, 0.81–0.85], logit [$P_{DNA}$] AUC: 0.67 [95% CI, 0.64–0.69], logit [$P_{M. pneumoniae}$] AUC: 0.75 [95% CI, 0.70–0.77], logit [$P_{G-}$] AUC: 0.68 [95% CI, 0.65–0.70], and logit [$P_{G+}$]: AUC 0.73 [95% CI, 0.70–0.75]). The combination of CRP, eosinophil, and either lymphocyte (AUC: 0.83) or HGB (AUC: 0.80 and 0.81) levels offer excellent ability to identify RNA virus infection and distinguish between bacterial and viral infections. The ROC curve showed that logit ($P_{infection}$) (AUC: 0.90, 0.88–0.92) had outstanding discrimination ability for the assessment of acute infection.

Using CRP level alone, the AUC value for predicting acute infections was 0.84 (95% CI, 0.79–0.86), which was greater than that for logit (*P*) (AUC: 0.75, 0.72–0.78) but lower than that of logit ($P_{infection}$) (AUC: 0.90, 0.88–0.92; p = 0.00). Using a combination of eosinophil, lymphocyte, and CRP levels, logit ($P_{RNA}$) showed significantly greater diagnostic accuracy (AUC: 0.83, 0.81–0.85) compared with the best single discriminatory marker (i.e., lymphocyte count), which had the greatest accuracy for predicting RNA virus infection (AUC: 0.77, 0.74–0.79). The combination of eosinophil and lymphocyte counts showed the best diagnostic accuracy, with AUC, sensitivity, and specificity of 0.67, 62.6%, and 67.4%, respectively, for predicting DNA virus infection. Although the AUC of the combination was < 0.70, it was statistically greater than that of eosinophil and lymphocyte counts alone (0.67 [95% CI, 0.64–0.69] vs. 0.60 [95% CI, 0.57–0.65] and 0.52 [95% CI, 0.49–0.55], respectively; p = 0.00 for both). Logit ($P_{G+}$) had significantly higher diagnostic accuracy than HGB and CRP levels (AUC: 0.77 [95% CI, 0.68–0.87] vs. 0.65 [95% CI, 0.60–0.67] and 0.63 [95% CI, 0.60–0.65], respectively). Compared with the combination of HGB, monocyte level, and NLR, logit ($P_{M. pneumoniae}$) had significantly better diagnostic accuracy than logit (*P*) for *M. pneumoniae* infection. The area under the ROC curve (AUROC) value was significantly higher for logit ($P_{M. pneumoniae}$) than monocyte count alone, which had the greatest accuracy for predicting *M. pneumoniae* infection (AUC: 0.75 [95% CI, 0.70–0.77] vs. 0.65 [95% CI, 0.62–0.68], respectively; p = 0.00). The AUC value for logit ($P_{G-}$) was < 0.70 and we failed to construct better models to diagnose DNA viral and G− organism infections (Table 2).

As shown in Table 3, using the cutoff value, the sensitivity, specificity, positive predictive value (PPV), and negative predictive value (NPV) of logit ($P_{infection}$) for the diagnosis of acute infection were 26.50, 80.9%, 85.7%, 98.3%, and 31.4%, respectively. Patients with a logit ($P_{infection}$) value of ≥ 26.50 mostly had infections, irrespective of the pathogen (PPV was 98.3% and specificity was 85.7%). The best cutoff values of logit ($P_{virus/bacteria}$) to diagnose viral and bacterial infections were −0.30 and −0.45, respectively, with a sensitivity of 70.2% and 70.7%, specificity of 78.7% and 78.9%, PPV of 64.4% and 70.8%, and NPV of 74.9% and 71.6%, respectively. When logit ($P_{virus/bacteria}$) was ≤ −0.30, the NPV to exclude a viral infection was 74.9%, while the PPV for the diagnosis of viral infection was 62.3% with a logit ($P_{virus/bacteria}$) > −0.3. When the score was ≤ −0.45, the PPV for the diagnosis of bacterial infection was 70.8% (suggesting that antibiotic therapy would be required); and the NPV to exclude a bacterial infection was 76.0% with a logit ($P_{virus/bacteria}$) > −0.45.

When logit ($P_{RNA}$) was ≥ −0.60, the PPV for the diagnosis of RNA virus infection was 81.3%; importantly, 68.9% of the total patients in the RNA virus group had IAV infection, suggesting that oseltamivir may be administered to these patients. Logit ($P_{G+}$) > −1.00 discriminated patients with G+ organism infection from other patients with sensitivity and specificity

**Table 2. AUROC values of significant parameters, logit ($P$), and logit ($P_{\text{control/virus/bacteria/RNA/DNA/M. pneumoniae/G–/G+}}$).**

| | CRP | WBC | HGB | Eosinophil | Monocyte | Lymphocyte | Logit ($P$) | Logit ($P_{\text{infection/virus/bacteria/RNA/DNA/M. pneumoniae/G–/G+}}$) | $P_1$**/ $P_2$*** |
|---|---|---|---|---|---|---|---|---|---|
| Acute infections | 0.84* (0.82–0.86) | 0.51(0.48–0.53) | 0.73 (0.68–0.77) | 0.76 (0.73–0.79) | 0.53 (0.47–0.58) | 0.75 (0.71–0.78) | 0.75 (0.72–0.78) | 0.90 (0.88–0.92) | 0.00/ 0.00 |
| Viruses | 0.53 (0.50–0.56) | 0.60(0.59–0.62) | 0.65 (0.63–0.67) | 0.72* (0.68–0.74) | 0.51 (0.49–0.54) | 0.70 (0.69–0.73) | 0.71 (0.69–0.73) | 0.80 (0.78–0.82) | 0.00/ 0.00 |
| RNA viruses | 0.59 (0.57–0.62) | 0.73(0.63–0.86) | 0.67 (0.65–0.69) | 0.72 (0.69–0.74) | 0.55 (0.52–0.57) | 0.77* (0.75–0.80) | 0.76 (0.73–0.78) | 0.83 (0.81–0.85) | 0.00/ 0.01 |
| DNA viruses | 0.59 (0.56–0.62) | 0.63(0.61–0.65) | 0.53 (0.50–0.56) | 0.60* (0.57–0.63) | 0.55 (0.52–0.59) | 0.52 (0.49–0.55) | 0.52 (0.47–0.54) | 0.67 (0.64–0.69) | 0.00/ 0.00 |
| Bacteria | 0.60 (0.58–0.63) | 0.62(0.60–0.65) | 0.72* (0.70–0.74) | 0.65 (0.63–0.68) | 0.60 (0.58–0.62) | 0.66 (0.64–0.68) | 0.75 (0.73–0.77) | 0.81 (0.79–0.82) | 0.01/ 0.01 |
| G– | 0.54 (0.51–0.57) | 0.60(0.58–0.62) | 0.65* (0.62–0.68) | 0.61 (0.58–0.63) | 0.58 (0.55–0.60) | 0.59 (0.58–0.64) | 0.65 (0.63–0.67) | 0.68 (0.65–0.70) | 0.28/ 0.16 |
| G+ | 0.63 (0.59–0.66) | 0.73(0.71–0.75) | 0.67* (0.64–0.69) | 0.61 (0.58–0.64) | 0.57 (0.53–0.60) | 0.62 (0.59–0.65) | 0.72 (0.69–0.74) | 0.73 (0.70–0.75) | 0.00/ 0.27 |
| *M. pneumoniae* | 0.63 (0.60–0.66) | 0.74(0.72–0.76) | 0.63 (0.60–0.65) | 0.61 (0.58–0.64) | 0.65* (0.62–0.68) | 0.57 (0.54–0.60) | 0.60 (0.57–0.62) | 0.75 (0.70–0.77) | 0.00/ 0.00 |

*The best single discriminatory marker for each group

**P value for logit ($P_{\text{control/virus/bacteria/RNA/DNA/M. pneumoniae/G–/G+}}$) compared with the best single discriminatory marker

*** P value for logit ($P_{\text{control/virus/bacteria/RNA/DNA/M. pneumoniae/G–/G+}}$) compared with logit ($P$). Abbreviations: AUROC: area under the receiver operating characteristic curve; CRP, C-reactive protein; HGB, hemoglobin; *M. pneumoniae*, *Mycoplasma pneumoniae*; G–, gram-negative organisms; G+, gram-positive organisms.

of 68.2% and 74.1%, respectively. This would allow the detection of more than half of patients with G+ organism infection with < 26% of false positives. Patients with logit ($P_{G+}$) value < −1.00 and logit ($P_{M.\ pneumoniae}$) value < −1.65 were not likely to have G+ or *M. pneumoniae* infection (NPVs: 90.2% and 92.2%, respectively) (Table 3). Because the reference ranges of full blood counts vary between neonates and older children, we also validated the models separately in neonatal patients. The sample sizes of neonates with viral and *M. pneumoniae* infections were inadequate to determine the diagnostic accuracy of these subsets in neonates; however, the AUC of logit ($P_{M.\ pneumoniae}$) (AUC: 0.86 [95% CI, 0.81–0.92]) was excellent (Tables 1–3). Similar to the overall study population, the AUC values for the DNA virus and G − organism infections in neonates were low (AUC: 0.64 [95% CI, 0.46–0.72], 0.55 [95% CI, 0.48–0.63]), indicating an unacceptable classification quality. The diagnostic utility of logit ($P_{\text{virus/bacteria}}$) for viral and bacterial infections and logit ($P_{G−}$) in neonates was low (AUC: 0.62 [95% CI, 0.43–0.68], 0.63 [95% CI, 0.56–0.75], and 0.55 (95% CI, 0.4–0.63), respectively]. Nonetheless, it had a good performance in distinguishing neonates with and without infections, including G+ bacterial infection (AUCs: 0.93 [95% CI, 0.89–0.95], 0.76 [95% CI, 0.65–0.80], respectively). Logit ($P_{\text{infection}}$) and logit ($P_{G+}$) had no discriminatory utility for neonates with p > 0.05 and a similar cutoff value was used for the overall study populations (Table 3).

For patients with logit ($P_{\text{infection}}$) value ≥ 26.50, logit ($P_{\text{virus/bacteria}}$) should be calculated; value < −0.45 indicates bacterial infection and value > −0.30 indicates viral infection. Then, logit ($P_{M.\ pneumoniae}$) should be calculated; value > −1.65 indicates *M. pneumoniae* infection. The results from the various models are not invariably mutually exclusive, such as the values of logit ($P_{\text{virus/bacteria}}$) between −0.45 and −0.30 indicated neither bacterial nor viral infection, since these formulae were not absolutely exact. However, the calculation of logit ($P$) would assist in the differential diagnosis of childhood infections. The logistic regression models for the different pathogens are presented in Fig 3A, whereas the algorithm of the suggested use of the model in routine clinical practice is presented in Fig 3B.

**Table 3. Diagnostic accuracy, sensitivity, and specificity of logit ($P_{\text{control/virus/bacteria/RNA/DNA/M. pneumoniae/G-/G+}}$) cutoffs for the overall study population and neonates.**

| Parameter | Parameter with largest AUCs | Cutoff value | | AUC | | 95% CI | | $P^*$ | Sensitivity (%) | | Specificity (%) | | PPV (%) | | NPV (%) | |
|---|---|---|---|---|---|---|---|---|---|---|---|---|---|---|---|---|
| | | All children | Neonates | All children | Neonates | All children | Neonates | | All children | Neonates | All children | Neonates | All children | Neonates | All children | Neonates |
| Acute infections | Logit ($P_{\text{infection}}$) | 26.50 | 26.76 | 0.90 | 0.93 | 0.88–0.92 | 0.89–0.95 | 1.99 | 80.90 | 83.90 | 85.70 | 98.50 | 98.30 | 80.50 | 31.40 | 95.90 |
| Viruses | Logit ($P_{\text{virus/bacteria}}$) | −0.30 | −1.80 | 0.80 | 0.62 | 0.78–0.82 | 0.43–0.68 | 0.15 | 70.20 | 64.70 | 78.70 | 58.90 | 64.40 | 20.80 | 74.90 | 98.30 |
| RNA viruses | Logit ($P_{\text{RNA}}$) | −0.60 | −4.10 | 0.83 | 0.59 | 0.81–0.85 | 0.37–0.78 | 0.17 | 70.30 | 66.70 | 83.70 | 32.10 | 81.30 | 13.30 | 75.60 | 98.60 |
| DNA viruses | Logit ($P_{\text{DNA}}$) | −1.75 | −1.80 | 0.67 | 0.64 | 0.64–0.69 | 0.46–0.72 | 0.63 | 62.60 | 72.20 | 67.40 | 46.40 | 30.50 | 29.30 | 87.80 | 89.20 |
| Bacteria | Logit ($P_{\text{virus/bacteria}}$) | −0.45 | −0.35 | 0.81 | 0.63 | 0.79–0.82 | 0.56–0.75 | 0.00 | 70.70 | 69.90 | 78.90 | 72.60 | 70.80 | 17.10 | 71.60 | 91.10 |
| G− | Logit ($P_{\text{G-}}$) | −1.40 | −0.35 | 0.68 | 0.55 | 0.65–0.70 | 0.48–0.63 | 0.00 | 67.90 | 42.50 | 70.20 | 66.70 | 45.20 | 67.80 | 83.20 | 39.30 |
| G+ | Logit ($P_{\text{G+}}$) | −1.00 | −0.78 | 0.73 | 0.76 | 0.70–0.75 | 0.65–0.80 | 1.18 | 68.20 | 71.90 | 74.10 | 67.30 | 38.60 | 48.20 | 90.20 | 84.10 |
| M. pneumoniae | Logit ($P_{\text{M. pneumoniae}}$) | −1.65 | −1.60 | 0.75 | 0.86 | 0.70–0.77 | 0.81–0.92 | 2.00 | 78.20 | 83.30 | 65.90 | 81.80 | 33.40 | 13.30 | 92.20 | 93.30 |

\* P value for AUCs of logit ($P_{\text{infection/virus/bacteria/RNA/DNA/M. pneumoniae/G-/G+}}$) for the entire study population compared with neonates. Abbreviations: AUC, area under curve; PPV, positive predictive value; NPV, negative predictive value; M. pneumoniae, Mycoplasma pneumoniae; G−, gram-negative organisms; G+, gram-positive organisms.

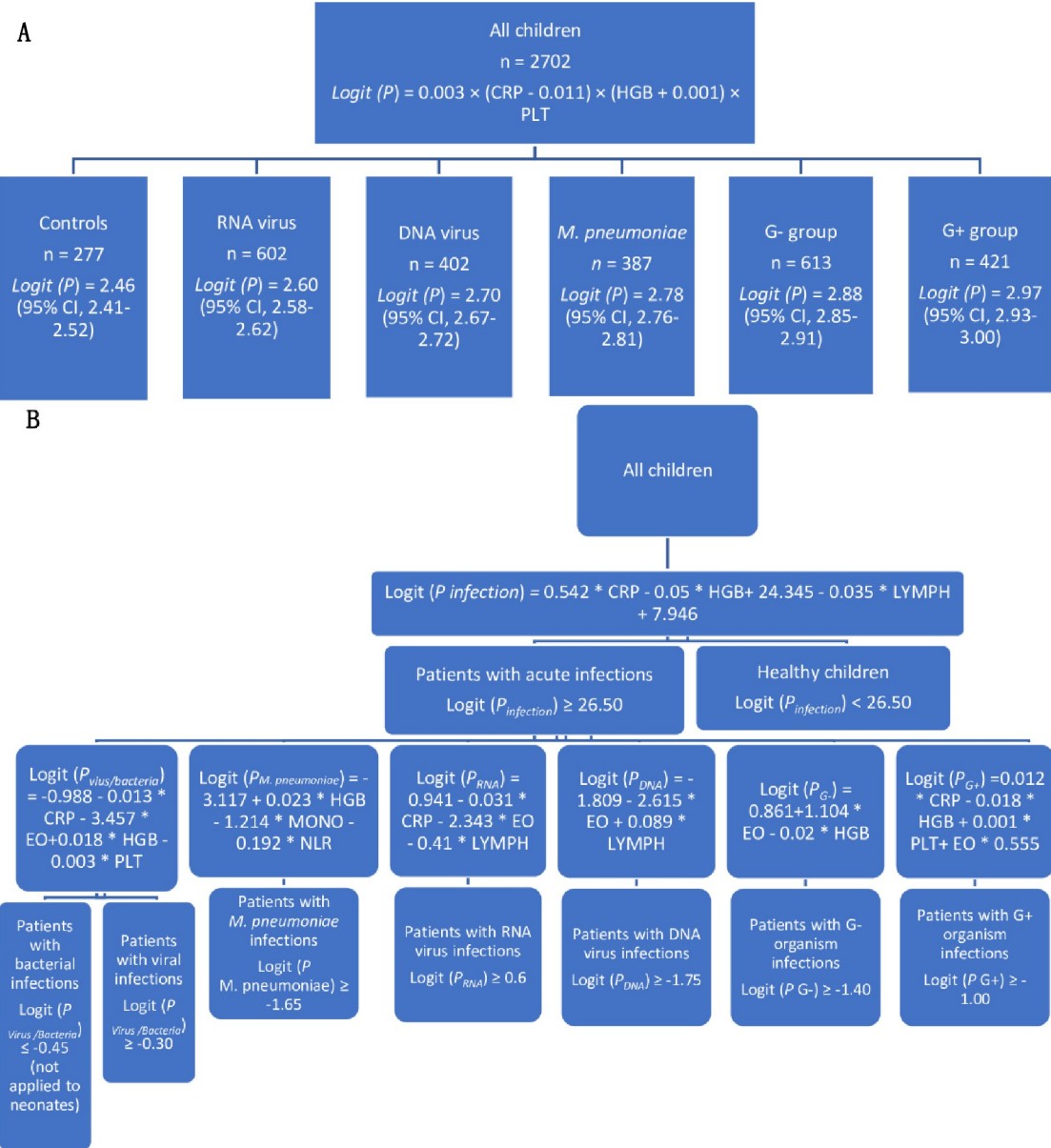

**Fig 3.** A, Logistic regression-based model for distinguishing among the six groups. B, Flow chart of the recommended used of the formulae in routine clinical practice.

## Discussion

This was the first study to demonstrate that RBC count, monocyte count, lymphocyte count, and eosinophil count did not perform well in distinguishing between the subsets of pathogens in infected individuals. The revised parameters developed in this study showed that these formulas had better accuracy than individual parameters with the largest AUROCs for the diagnosis of different pathogen types. The variables used in these models are widely used in clinical practice and easily available. The usefulness of marker combinations for distinguishing between *M. pneumoniae*, RNA virus, DNA virus, G− organism, and G+ organism infections has not been studied previously. These marker combinations may be better at guiding medication selection than dividing patients into viral and bacterial infection groups. Appropriate

antiviral treatment and antibiotic selection requires timely determination of whether the infection is caused by DNA or RNA viruses, or G− or G+ bacteria. These marker combinations may be useful for the early diagnosis and improved outcomes of infections in pediatric patients.

Neonatal infections are particularly difficult to diagnose and no reliable predictors exist. Failure to identify acute infection may lead to delayed initiation of therapy and severe illness. Thus, the identification of predictors of neonatal infection is important. History and physical examination do not reliably exclude acute infections in neonates. Logit ($P_{infection}$) had a PPV of 98.3% to predict acute infection, which allows the appropriate diagnosis and empirical antibiotic therapy.

The WBC count is increased in bacterial infections [9]. However, the total and differential WBC counts are also affected by clonal myeloid disorders as well as immune and inflammatory conditions. In line with previous prospective studies in children with infections [8], the WBC count had an AUC of < 0.70 and was a weak predictor of bacterial and viral infections in children (Table 2). The bacterial infection group had lower HGB and higher CRP levels, even after adjusting for patient's age, consistent with the results of previous studies [8, 9]. Ballin et al. demonstrated that bacteremia is accompanied by a significant decrease in HGB level in children without evidence of hemolytic anemia [26]. In addition, the serum iron level is a strong predictor of disease outcome in intensive care unit patients [27].

It is difficult to differentiate between DNA virus and bacterial infections because the laboratory parameters were similar between them. For example, elevated CRP concentration was also noted in pediatric adenovirus patients in the absence of secondary bacterial infection as well as in patients with bacterial infection, indicating that adenoviruses trigger an immediate inflammatory host response resembling that triggered by invasive bacterial infection [28]. In this study, the eosinophil count in patients with infections was significantly lower than that in controls.

The blood WBC count varies with age, with higher counts in infants and toddlers compared with adolescents and adults [29]. Thus, the identification of predictors of neonatal sepsis is important. Nonetheless, logit ($P_{virus/bacteria}$) and logit ($P_{G−}$) were not useful for neonates. However, logit ($P_{infection}$) is an excellent predictor of acute infection and logit ($P_{G+}$) can improve the diagnostic efficiency in neonates.

Several limitations of our study should be acknowledged. First, the various models constructed in this study are not invariably mutually exclusive. Second, although we adjusted for several potential confounding factors, the possibility of the effect of residual confounding factors on risk factor analysis cannot be excluded. Third, the cohorts were categorized based on the laboratory results only. The performance of these tests suggests that the clinicians suspected a viral or bacterial infection so the sample population may be biased. Fourth, the results are not applicable to patients with multi-organism infections, cancer patients, or those with underlying inflammatory conditions. Fifth, this was a retrospective study; therefore, prospective validation of the models is required. Finally, the pathological course of the disease is unknown, i.e., the time from disease onset, which precedes hospitalization. Understanding the changes in blood parameters measured at disease onset and subsequently thereafter can help to make the accurate diagnosis. Presumably, days 3–7 after symptom onset may be ideal for testing blood parameters. However, this information requires further confirmation in prospective studies.

## Conclusions

The combination of frequently tested peripheral blood parameters (such as CRP, HGB, eosinophil, monocyte, and lymphocyte levels) can differentiate between children with and without acute infection, and provide a relatively sensitive and specific indication of the infection type.

## Author Contributions

**Conceptualization:** Weiying Wang.

**Data curation:** Weiying Wang.

**Formal analysis:** Shu Hua Li.

**Writing – original draft:** Weiying Wang.

**Writing – review & editing:** Shu Hua Li.

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
