## [Decision Letter · Decision Letter 0]

17 Jan 2022

PONE-D-21-32536Differential diagnosis of computational methods with peripheral blood parameters in children with certain type of infection.PLOS ONE

Dear Dr. Wang,

Thank you for submitting your manuscript to PLOS ONE. After careful consideration, we feel that it has merit but does not fully meet PLOS ONE’s publication criteria as it currently stands. Therefore, we invite you to submit a revised version of the manuscript that addresses the points raised during the review process.

The reviewers both felt that the manuscript had merit but both felt that there were a number of grammatical and typographical errors which required correction. This included a number of abbreviations which have not been defined. In addition, reviewer 1 raised some more substantive concerns. These include: 1. The clarification of the groups which were included in the analysis - for example those children with "contaminated" cultures. In addition, there is reference in the discussion to the utility of the models in neonates but neonatal blood counts are different from older population groups and should be independently assessed 2. All 3 tables were cut-off and table 2 and 3 should be reordered. 3. Some information included in the discussion is not adequately shown in the results section

We look forward to receiving your revised manuscript.

Kind regards,

Elizabeth S. Mayne, M.D.

Academic Editor

PLOS ONE

Journal Requirements:

2. Thank you for submitting the above manuscript to PLOS ONE. During our internal evaluation of the manuscript, we found significant text overlap between your submission and the following previously published works, some of which you are an author.

- https://pubmed.ncbi.nlm.nih.gov/32411832/

- https://bmjpaedsopen.bmj.com/content/4/1/e000640.full

- https://link.springer.com/article/10.1007/s15010-019-01383-6?code=20d8c7b6-72e0-4ad7-a307-f9cc834c5dc8&error=cookies_not_supported

- https://linkinghub.elsevier.com/retrieve/pii/S1875957218300275

- https://pubmed.ncbi.nlm.nih.gov/30443990/

- https://www.wjgnet.com/1007-9327/full/v26/i32/4857.htm

- https://www.pediatr-neonatol.com/article/S1875-9572(18)30027-5/fulltext

- https://journals.sagepub.com/doi/10.1177/0961203319827646

Please revise the manuscript to rephrase the duplicated text, cite your sources, and provide details as to how the current manuscript advances on previous work. Please note that further consideration is dependent on the submission of a manuscript that addresses these concerns about the overlap in text with published work.

“None declared.”

6. Please include your tables as part of your main manuscript and remove the individual files. Please note that supplementary tables (should remain/ be uploaded) as separate "supporting information" files.

Additional Editor Comments:

The reviewers both felt that the manuscript had merit but both felt that there were a number of grammatical and typographical errors which required correction. This included a number of abbreviations which have not been defined. In addition, reviewer 1 raised some more substantive concerns. These include:

1. The clarification of the groups which were included in the analysis - for example those children with "contaminated" cultures. In addition, there is reference in the discussion to the utility of the models in neonates but neonatal blood counts are different from older population groups and should be independently assessed

2. All 3 tables were cut-off and table 2 and 3 should be reordered.

3. Some information included in the discussion is not adequately shown in the results section

Reviewers' comments:

Reviewer's Responses to Questions

**Comments to the Author**

1. Is the manuscript technically sound, and do the data support the conclusions?

Reviewer #1: Partly

Reviewer #2: Yes

2. Has the statistical analysis been performed appropriately and rigorously? 

Reviewer #1: I Don't Know

Reviewer #2: Yes

3. Have the authors made all data underlying the findings in their manuscript fully available?

Reviewer #1: No

Reviewer #2: Yes

4. Is the manuscript presented in an intelligible fashion and written in standard English?

Reviewer #1: No

Reviewer #2: No

5. Review Comments to the Author

Reviewer #1: This study has assessed the value of including multiple laboratory parameters in a model in order to differentiate children with and without infection, as well as to determine the likely aetiopathogen. Although cumbersome, the models have shown fair diagnostic performance in this retrospective cohort. However, the manuscript requires substantial editing/clarification.

1) The Title does not clearly convey the intent or findings of the study and should be edited.

2) The abstract requires extensive editing:

a. The introduction is too brief, and doesn’t clearly relay the rationale for the study.

b. The methods are too vague (particularly regarding the computational analysis performed).

c. The results are confusing and need to be re-written.

d. The conclusion should be edited to state that it is the combination of these blood parameters that are beneficial (as it stands it suggests that they are individually useful).

e. The abbreviation “MP” has not been defined.

f. The abbreviations for Gram Positive (G+) and negative (G-) have not been introduced, but have been used throughout the rest of the manuscript.

g. The country the study has been performed in should be stated.

3) In the introduction, the following statement: “lower concentrations of CRP are indicative of both bacterial and viral infections.” is not correct. Do you perhaps mean that lower concentrations of CRP can occur in both bacterial and viral infections?

4) In the introduction, the following statement: “This highlights the patients presenting with low CRP levels as a group with the most uncertainty.” is disputable, since a high CRP can also occur in the absence of infection (for instance in inflammatory or malignant conditions).

5) What criteria were used for including children as controls?

6) Please clarify if patients with contaminated cultures were excluded or included as controls? It states under the exclusion criteria that these were counted as having negative cultures.

7) Under Laboratory data, the meaning of the following sentence “Hence, patient had blood counts combined with CRP testing available that were drawn between 3 to 7 days after symptom onset.” is unclear. Do you mean that you retrieved/recorded full blood count and CRP results collected between 3 and 7 days from symptom onset?

8) All 3 of the Tables have been cut off. Please adjust so that they are visible in their entirety.

9) On the last page of the results, the following is stated: “When the score was equal to or less than -0.45, the PPV in the diagnosis of bacterial infection was 70.8%, suggesting that anti-infective therapy was required.” Should this not read: “When the score was equal to or greater than -0.45, the PPV in the diagnosis of bacterial infection was 70.8%, suggesting that anti-infective therapy was required”?

10) Were the results from the various models invariably mutually exclusive? In other words, did results from the different analyses ever indicate both bacterial and viral infection? If so, this would considerably ameliorate the clinical utility of these formulae.

11) It would be helpful to include a flow diagram of how the authors suggest these formulae would be used in routine clinical practice. Eg Begin with the Logit (Pcontrol) formula, If value >x proceed to Logit (Pbacterial/viral) formula, etc.

12) In the Discussion, the utility of this approach is discussed among neonatal patients. However, the number of neonates included in this study has not been specified in the results, and since neonatal full blood count results can be very different from those seen in older children, the models should be validated in neonatal patients separately before any conclusions about the utility of these findings can be made in this patient subgroup.

13) In the discussion, the following statement: “The relation of WBC levels and infection remains controversial as well as neutrophils” is unsubstantiated. The authors have already stated that the white cell count is well known to be elevated in bacterial infection.

14) There are several parts of the discussion which seem irrelevant. These include the following:

“Hepcidin is a key regulator of iron homeostasis.24 Inflammation-induced hepcidin interacts with ferroportin, which becomes internalized and degraded and ultimately leads to intracellular iron sequestration and decreased iron absorption in the duodeum.2522 Furthermore, hepcidin-induced low iron levels were related to both the long-term and short-term survival rates of critically ill individuals.23” and

“Lymphocytes play important roles in defense against and recovery from multiple virus infections especially retroviruses infections, demonstrated by studies using adoptive transfer or host immunosuppression.27-28 MP pneumonia is considered to be in part attributed to immune-mediated responses in which monocytes and its subsets appear to be important element 29 . Blood platelets were presented as active players in antimicrobial host defense and the induction of inflammation and tissue repair in addition to their participation in hemostasis via releasing the content of their alpha-granules, which include an arsenal of bioactive peptides, such as CC-chemokines and CXC-chemokines and growth factors for endothelial cells, smooth muscle cells and fibroblasts.30-31”

Please clarify or remove these sections.

15) In the discussion, it is stated that “CRP levels were independent of the duration of illness, indicating that adenoviruses trigger an immediate inflammatory host response resembling invasive bacterial infection.25” This has not been shown in the Results. Please include it.

16) In the Discussion, it is stated that “In this study, the number of eosinophils in the infected group was significantly lower than that of controls, while the percentage of eosinophils was similar in the two groups,”. Please clarify which 2 groups you are referring to.

17) In the same sentence, it is stated “suggesting that eosinophil depletion may also be the cause of infection”. This is incorrect. Do you mean that eosinophil depletion may be caused by infection?

18) Additional limitations to this study include the fact that the results are not applicable to patients with mulit-organism pathology, in cancer patients or in those with underlying inflammatory conditions.

19) The article contains numerous minor grammatical errors. Some, but not all, have been corrected below. The manuscript would benefit from English language editing.

20) The data for the study has not been provided.

Minor Corrections:

1) The Introduction has no heading

2) In the introduction, the abbreviation “FN” should be defined.

3) In the introduction, the abbreviation proADM should be spelled out/defined when first used.

4) In the introduction, the following sentence: “It is unfortunate that almost three-quarters of all patients thought to have a purely viral syndrome received treatment with antibiotics.” should be referenced.

5) In the introduction, the following statement: “However, reliable physical examination findings and routinely individual laboratory investigations are not currently available to help clinicians differentiate benign viral infections or a case of over-swaddling from serious bacterial infections in children.” suggests that over-swaddling is caused by bacterial infection. A suggested rephrasing of this sentence is as follows: “However, physical examination findings and routine laboratory investigations are not able to help clinicians accurately differentiate benign viral infections from over-swaddling or serious bacterial infections in febrile children.”

6) In the Introduction, the following sentence: “The aim of this study was to identified optimal，commonly available parameters in the differential diagnosis of RNA, DNA viral, MP, G- and G+ infection, which has not been previously studied in these groups and to determine cut-off values that could aid clinicians in the evaluation of febrile pediatric patients.” is unclear. A suggested rephrasing is as follows: “The aim of this study was to assess commonly available blood parameters in differentiating RNA viral, DNA viral, MP, G- and G+ infection, and to determine cut-off values that could aid clinicians in the evaluation of febrile pediatric patients.”.

7) Under the exclusion criteria and in Figure 1, change “cross-contamination” to “multi-organism infection” or “dual pathology”.

8) In the methods, 2 sentences have been started with a number in numerical format. These numbers should be written out.

9) Under the Study Population, the abbreviation ED should be spelled out.

10) All of the organisms abbreviated in Figure 2 should be listed in the Legend.

11) In the results, the meaning of the following sentence: “Because the diseases have different age prevalence, age is considered the most important demographic characteristic” is not clear.

12) In Table 1, please include the interquartile range for the age (not the 95% CI).

13) Table 2 should be placed before Table 3 or they should be renamed accordingly.

14) In the Discussion, the following sentence: “However, total and differential WBC are also associated with medullar, immune and inflammatory disorders.” should be rephrased as follows: “However, total and differential WBC counts are also affected by clonal myeloid disorders as well as immune and inflammatory conditions”.

15) The Conclusion is poorly phrased. A suggested rephrasing is as follows: “The proposed approach for using the computational method, combining frequently tested peripheral blood parameters such as CRP, HGB, eosinophil, monocyte and lymphocyte counts, can assist in differentiating between children with and without acute infection and provide a relatively sensitive and specific indication of the type of infection present.”

Reviewer #2: The manuscript is scientifically sound, however requires grammatical review and update by an English scientific writer.

I have made a number of comments and edits, however these are not comprehensive.

6. PLOS authors have the option to publish the peer review history of their article (what does this mean?). If published, this will include your full peer review and any attached files.

Reviewer #1: No

Reviewer #2: **Yes: **Jessica June Sancroft Opie

---

## [Author Response · Author response to Decision Letter 0]

1 Apr 2022

Details are responded in the letter to reviewers.

---

## [Decision Letter · Decision Letter 1]

15 Jun 2022

PONE-D-21-32536R1Use of common blood parameters for the differential diagnosis of childhood infectionsPLOS ONE

Dear Dr. Wang,

Thank you for submitting your manuscript to PLOS ONE. After careful consideration, we feel that it has merit but does not fully meet PLOS ONE’s publication criteria as it currently stands. Therefore, we invite you to submit a revised version of the manuscript that addresses the points raised during the review process.

Both reviewers feel that the manuscript is substantially improved, however, there are still a number of minor errors. Specifically, these relate to the description of the statistical analysis and these must be corrected.

We look forward to receiving your revised manuscript.

Kind regards,

Elizabeth S. Mayne, M.D.

Academic Editor

PLOS ONE

Journal Requirements:

Additional Editor Comments (if provided):

Both reviewers feel that the manuscript is substantially improved, however, there are still a number of minor errors. Specifically, these relate to the description of the statistical analysis and these must be corrected.

Reviewers' comments:

Reviewer's Responses to Questions

**Comments to the Author**

1. If the authors have adequately addressed your comments raised in a previous round of review and you feel that this manuscript is now acceptable for publication, you may indicate that here to bypass the “Comments to the Author” section, enter your conflict of interest statement in the “Confidential to Editor” section, and submit your "Accept" recommendation.

Reviewer #1: (No Response)

Reviewer #2: (No Response)

2. Is the manuscript technically sound, and do the data support the conclusions?

Reviewer #1: Partly

Reviewer #2: Yes

3. Has the statistical analysis been performed appropriately and rigorously? 

Reviewer #1: I Don't Know

Reviewer #2: Yes

4. Have the authors made all data underlying the findings in their manuscript fully available?

Reviewer #1: No

Reviewer #2: Yes

5. Is the manuscript presented in an intelligible fashion and written in standard English?

Reviewer #1: Yes

Reviewer #2: Yes

6. Review Comments to the Author

Reviewer #1: This revised manuscript shows substantial improvement as compared to the previous submission. However, there are still some queries/weaknesses which need to be addressed:

1) The background of the Abstract is still weak.

2) What is the relevance of the data presented regarding the month of infection? (line 159 and Fig 2)

3) There is an error in Lines 263-265. This should read as follows: “When logit (Pvirus/bacteria) was ≤ −0.30, the NPV to exclude a viral infection was 74.9%, while the PPV for the diagnosis of viral infection was 62.3% with a logit (Pvirus/bacteria) >-0.3. When the score was ≤ −0.45, the PPV for the diagnosis of bacterial infection was 70.8% (suggesting that antibiotic therapy would be required) and the NPV to exclude bacterial infection was 76.0% with a logit (Pvirus/bacteria) >-0.45.” This sentence was better expressed in the original manuscript.

4) The sentence in lines 273-275 should be rephrased as follows: “Since the reference ranges of full blood counts vary between neonates and older children, we also validated the models separately in neonatal patients.”

5) The 1st sentence of the discussion is not well substantiated by the data presented. For example, the AUC data was not presented for the Neuts, WBC or NLR, and there were significant differences in the Neuts and NLR between the groups in Table 1.

6) In line 299, reference is made to the Neutrophil% and the Lymphocyte%, but the data presented appears to be absolute Neutrophil and lymphocyte counts (Table 1).

7) What is the basis for stating that the WBC count was the weakest predictor of pathogens among children with infection compared to the other parameters (lines 320-322)? The WBC was significantly different between the groups in Table 1, unlike the RCC, the eosinophils count and the monocyte count…

8) The following statement in the discussion “In this study, the eosinophil count in patients with infections was significantly lower than that in controls, while the percentage of eosinophils was similar between children with and without infections.” (lines 332-334) is not supported by the data presented.

9) Lines 336-341 are almost a word for word repeat of lines 312-316.

10) Include in the limitations that prospective validation of the models is also required.

11) In Table 1, an R-value is presented (presumably a correlation co-efficient)? How this value has been derived is not clear?

12) In Fig 3b, should the Logit (PRNA) and the Logit (PDNA) not only be performed in the cases with a Logit (Pvirus/bacteria) >-0.3, and the Logit (P M.pnemonia), the Logit (PG+) and the Logit (PG-) only in the cases with a Logit (Pvirus/bacteria) ≤-0.45? Is it your intention for all of the models to be applied to all children?

Reviewer #2: Dear Authors

Thank you, the revised version is much improved.

However, some minor corrections remain, with clarification needed in some areas, including the inclusion of some of your figures (which include months 1 - 12).

The attached document provides my detailed comments in pop up notes.

Kind regards

7. PLOS authors have the option to publish the peer review history of their article (what does this mean?). If published, this will include your full peer review and any attached files.

Reviewer #1: No

Reviewer #2: No

---

## [Author Response · Author response to Decision Letter 1]

13 Jul 2022

Dear Dr. Mayne and Reviewers,

Thank you for giving us the opportunity to correct the manuscript. We have reviewed the comments provided by the journal editors and reviewers. We have addressed the comments in the revised manuscript and the changes are highlighted.

Reviewer #1. Thank you for your constructive comments. Please see my responses to your comments below:

1) Abstract

We have added a sentence regarding the background.

2) Figure 2B–C can assist physicians to predict the prevalent pathogen in each month.

3–4) We have made additional revisions to the revised manuscript.

5) Although there were significant differences in the WBCs, neutrophils, and NLR between the groups (Table 1), their AUCs were < 0.70. We have added the AUC of WBCs in Table 2.

6) Our statement refers to the absolute neutrophil and lymphocyte counts, rather than the percentages of neutrophils and lymphocytes. We have clarified this in the manuscript.

7) The AUC of WBCs was < 0.70 (Table 2). The total and differential WBC counts are affected by stringent state as well as immune and inflammatory conditions.

8) The data are correct, but we have not presented the percentage of eosinophils. We have deleted the sentence.

9–10) The changes made to the revised manuscript are highlighted.

11) An R-value is presented as a correlation coefficient between the parameters and the groups using bivariate analysis.

12) In Figure 3b, the Logit (PRNA) and Logit (PDNA) were not only performed in cases with a Logit (Pvirus/bacteria) > −0.3; and Logit (PM. pneumonia), Logit (PG+), and Logit (PG−) were not only performed in cases with Logit (Pvirus/bacteria) ≤ −0.45. The model was developed to rapidly determine the pathogens using the formula after obtaining the blood routine examination results. The Logit (Pinfection), Logit (PRNA), Logit (PDNA), Logit (PG+), and Logit (PM. pneumoniae) should be used to classify all children (p > 0.05 for the comparison between neonates and other children; Table 3). Logit (Pvirus/bacteria) should not be used to classify neonates because of a p < 0.05 for the comparison between neonates and other children. 

Reviewer #2. We appreciate your positive comments. We have revised the manuscript for language and scientific content; the changes are highlighted in the revised manuscript. Figure 2B–C may assist physicians to predict the prevalent pathogen in each month.

Yours sincerely,

Wei-ying Wang

---

## [Decision Letter · Decision Letter 2]

5 Aug 2022

Use of common blood parameters for the differential diagnosis of childhood infections

PONE-D-21-32536R2

Dear Dr. Wang,

We’re pleased to inform you that your manuscript has been judged scientifically suitable for publication and will be formally accepted for publication once it meets all outstanding technical requirements.

Kind regards,

Elizabeth S. Mayne, M.D.

Academic Editor

PLOS ONE

Additional Editor Comments (optional):

Reviewers' comments:

Reviewer's Responses to Questions

**Comments to the Author**

1. If the authors have adequately addressed your comments raised in a previous round of review and you feel that this manuscript is now acceptable for publication, you may indicate that here to bypass the “Comments to the Author” section, enter your conflict of interest statement in the “Confidential to Editor” section, and submit your "Accept" recommendation.

Reviewer #1: (No Response)

Reviewer #2: All comments have been addressed

2. Is the manuscript technically sound, and do the data support the conclusions?

Reviewer #1: Yes

Reviewer #2: (No Response)

3. Has the statistical analysis been performed appropriately and rigorously? 

Reviewer #1: I Don't Know

Reviewer #2: (No Response)

4. Have the authors made all data underlying the findings in their manuscript fully available?

Reviewer #1: Yes

Reviewer #2: (No Response)

5. Is the manuscript presented in an intelligible fashion and written in standard English?

Reviewer #1: Yes

Reviewer #2: (No Response)

6. Review Comments to the Author

Reviewer #1: The manuscript has been substantially improved, and the authors have addressed all my concerns, It is now suitable for publication.

Reviewer #2: (No Response)

7. PLOS authors have the option to publish the peer review history of their article (what does this mean?). If published, this will include your full peer review and any attached files.

Reviewer #1: No

Reviewer #2: No

---

## [Editor Report · Acceptance letter]

2 Sep 2022

PONE-D-21-32536R2 

Use of common blood parameters for the differential diagnosis of childhood infections 

Dear Dr. Wang:

I'm pleased to inform you that your manuscript has been deemed suitable for publication in PLOS ONE. Congratulations! Your manuscript is now with our production department. 

Kind regards, 

on behalf of

Dr. Elizabeth S. Mayne 

Academic Editor

PLOS ONE